# BRCT Domains: Structure, Functions, and Implications in Disease—New Therapeutic Targets for Innovative Drug Discovery against Infections

**DOI:** 10.3390/pharmaceutics15071839

**Published:** 2023-06-27

**Authors:** José Peña-Guerrero, Celia Fernández-Rubio, Alfonso T. García-Sosa, Paul A. Nguewa

**Affiliations:** 1ISTUN Institute of Tropical Health, Department of Microbiology and Parasitology, University of Navarra, IdiSNA (Navarra Institute for Health Research), E-31008 Pamplona, Navarra, Spain; jpena.1@alumni.unav.es (J.P.-G.); cfdezrubio@unav.es (C.F.-R.); 2Chair of Molecular Technology, Institute of Chemistry, University of Tartu, Ravila 14a, 50411 Tartu, Estonia; alfonsog@ut.ee

**Keywords:** BRCT, infection, disease, cancer, leishmania, therapeutic target, drug

## Abstract

The search for new therapeutic targets and their implications in drug development remains an emerging scientific topic. BRCT-bearing proteins are found in Archaea, Bacteria, Eukarya, and viruses. They are traditionally involved in DNA repair, recombination, and cell cycle control. To carry out these functions, BRCT domains are able to interact with DNA and proteins. Moreover, such domains are also implicated in several pathogenic processes and malignancies including breast, ovarian, and lung cancer. Although these domains exhibit moderately conserved folding, their sequences show very low conservation. Interestingly, sequence variations among species are considered positive traits in the search for suitable therapeutic targets, since non-specific drug interactions might be reduced. These main characteristics of BRCT, as well as its critical implications in key biological processes in the cell, have prompted the study of these domains as therapeutic targets. This review explores the possible roles of BRCT domains as therapeutic targets for drug discovery. We describe their common structural features and relevant interactions and pathways, as well as their implications in pathologic processes. Drugs commonly used to target these domains are also presented. Finally, based on their structures, we describe new drug design possibilities using modern and innovative techniques.

## 1. Introduction

The precise understanding of drug-target interactions remains a valuable asset for numerous strategies such as the screening of drug candidates, drug repositioning, reduction of drug side-effects, and the discovery of novel protein-drug interactions [1]. Current estimates suggest that over 95% of presently known drug targets are proteins [2]. Therefore, there is an increasing interest in the identification of functional and structural characteristics that differentiate drug targets from non–drug targets [1]. Several scientific works have attempted to carry out this task. Early data focused on characteristics such as structural fold, protein family, associated pathways, tissue distribution, and chromosome location [3,4]. More modern approaches aim to study different features of the target such as protein sequence, hydrophobicity, charge, secondary structure, transmembrane regions, and post-translational modifications (PTMs), among others [5,6,7]. The aforementioned characteristics of new druggable molecules were studied in comparison with the currently annotated druggable proteins and with the rest of the proteins labelled as “non-drug targets”. However, many of the proteins labelled as “non-drug targets” could presently be druggable [8]. Consequently, the use of the set of currently known “drug targets” to represent the totality of the druggable proteome creates a bias that reduces the capability to identify novel sets of druggable proteins [1]. Typically, drug target proteins are composed of G-protein coupled and nuclear receptors, ion channels, and enzymes [9,10]. In contrast, recent studies advocate for the drug targeting of non-enzymes (scaffolding, regulatory, and structural proteins) and proteins participating in protein-protein interactions (PPIs) [9,10,11,12], thus increasing the number of potential drug targets. These data remain relevant considering that many of the existing drug targets remain undiscovered.

The protein encoded by the breast cancer susceptibility gene (*BRCA1*) contains at its C-terminal end two copies of a conserved 90–100 amino acids segment that was named BRCT (BRCA1 C-Terminus) domain. Over the years, BRCT domains have been identified in other proteins as well, located not only at C-terminal ends, but also in multiple or single copies [13]. Biochemical and sequence analyses suggest that BRCT domains mostly support protein-protein interactions and are typically linked with DNA repair, recombination, and cell cycle control [13,14,15]. Although there are BRCT domains present in Archaea, Bacteria, Eukarya, and even viruses, BRCT-containing proteins are more commonly observed in bacteria and eukaryotes [16], as it has been shown that the number of BRCT-bearing proteins within a genome increases according to the genome complexity [16].

This review explores the possible roles of BRCT domains as therapeutic targets for drug discovery. We start by describing the structural features of BRCT domains and analysing the interactions mediated by these domains. We also analyse the pathways in which these domains are implicated, taking into special consideration pathologic and infective processes. Lastly, we review the current main targets of these domains and the compounds designed to specifically inhibit them. Altogether, this review may contribute to a better understanding of BRCT domains and their possible implications for innovative drug development.

## 2. BRCT Domains: Structure and Architecture

The human BRCT domain was first resolved from the crystal structure of the N-terminal BRCT of the X-ray repair cross-complementing protein 1 (XRCC1), determined by X-ray crystallography to a 3.2 Å resolution. Its tertiary structure features a central core of four-stranded parallel β-sheet (β1, β2, β3, and β4) flanked by two α-helices (α1 and α3) on the C-terminal end, a single α-helix (α2) on the N-terminal end, and two surface loops connecting β1 with α1 and α2 with β3 (the overall structure being β1-α1-β2-β3-α2-β4-α3) (Figure 1).

Hydrophobic interactions are formed between residues from α1 with those from β1 and β2 as well as residues from α2 with those from β4 [17]. In addition, five conserved hydrophobic motifs, named A, B, C, D, and E, can be found in β1, α1 and α1-β2 loop, β3, α3, and C-terminal end, respectively [16,18].

Analyses of alignments between representative BRCT sequences reveal very low identity, despite the predicted conserved structure. According to the amino acid sequence, the conserved hydrophobic amino acid clusters are located within the central β-sheet, on α1 and α3, and at the N- and C-terminal region of the domain [13,18]. Moreover, other highly conserved residues within the family are a double Gly-Gly motif located in the short loop between α1 and β2, and those which form helix α3 surface and are involved in the interaction between α1 and α3, suggesting that this two-stranded helical bundle is a crucial component of the domain (Figure 1).

Variations in BRCT structure and sequence are mainly located in α2, which varies in both size and amino acid composition [17] and is even absent in some members such as DNA Ligase III BRCT [19]. Numerous modifications are also detected in the surface loops connecting sheets and helices. These variable regions are supposed to form stable protein-protein interaction surfaces [17]. In summary, the most conserved residues are located in central β-sheet, α1 and α3, while α2 is the most variable region within the BRCT family.

The phylogenetic relationships of the BRCT domain have been explored using PES BRCT as a model [20] (Figure 2). It is known that BRCT domains bear two motifs, I and II, and contain clusters of conserved residues. Motif I features a glycine-(glycine/arginine) doublet, while motif II includes a highly conserved tryptophan [13]. Both motifs are expected to be identifiable within the PES1 BRCT domain [21]. Although the highly conserved Gly-Gly position, which corresponds to the aforementioned motif I, is present in the alignment, the researchers were not able to identify the conserved tryptophan corresponding to motif II in trypanosomatids and *Plasmodium*, as it was substituted by the aromatic tyrosine. However, the presence of a conserved cluster of tyrosine-valine (short nonpolar amino acid), followed by a fully conserved core of glutamine-proline-glutamine and the previously mentioned tryptophan/tyrosine-leucine (short nonpolar amino acid), was found. Considering the aforementioned differences between human and trypanosomatid PES BRCT domains, as well as the gap in positions 35–45, the generation of selective LmjPES BRCT inhibitors that may not interact with human PES1 BRCT seems feasible [20].

In addition, the architecture of BRCT domains is highly variable. The top five common architectures are as follows: isolated; single domain + FHA (Forkhead associated); tandem domain + RING; two single domains + two homeodomain; and tandem domain + pleckstrin homology (PH) domain-like + Dbl homology (DH) domain [22]. Regarding the typical BRCT tandem—for example, the one found in BRCA1 protein—the domains fold in a head-to-tail orientation, engaging a wide hydrophobic interface composed of the α2 helix of the N-terminal repeat, which is packed against α1 and α3 of the C-terminal repeat. The conservation of α1 and α3 residues may indicate that this specific folding is shared by other proteins bearing this domain [23]. Nonetheless, there are variations of the typical tandem BRCT folding such as, for instance, the tandem BRCT domains of DNA Ligase IV (LigIV), which are separated by a significantly longer link [19].

BRCT domains have been associated with phosphopeptide-linking. A conserved phosphoserine-binding pocket which includes a Ser/Thr-Gly motif in the variable β1-α1 connecting loop (Ser1655 and Gly1656 of human BRCA1) and a Thr/Ser-X-Lys motif at the N terminus of α2 of the same repeat (residues 1700–1702 of human BRCA1) is present in those BRCTs predicted to have a phosphopeptide binding function [14]. In BRCA1, this structure creates a cleft in which the ligand (bearing a pSXXF motif) can be bound following a “two-anchor” interaction. pSer maintains polar interactions with conserved residues in β1 and α2, whereas +3 F sidechain inserts into a deep hydrophobic pocket of the interface between the first and second BRCT domain. Furthermore, the residues that coordinate the binding to pSer are highly conserved, in contrast to the residues that interact with pSer+3. Such variations of the residues interacting with pSer+3 may explain why different BRCT domains prefer different phosphopeptides, as well as their location at the pSer+3 position [24]. Additionally, residues located outside this pSXXF motif also play a role in this interaction, as different phosphopeptides sharing the same core DpSPVF sequence and peptide length, but bearing distinct N-terminal and C-terminal sequences, interact with BRCA1 tandem BRCT domains with significantly different affinities. Due to the higher number of hydrophobic interactions, those residues located in the N-terminal side contribute to the binding affinity [25].

Interactions with phosphopeptides have also been described in single BRCT, specifically the contact of phosphorylated Bloom syndrome RecQ-like helicase (BLM) with BRCT5 of murine Topoisomerase IIβ binding protein 1 (TopBP1) [26]. In this case, the phosphorylated serine of BLM may interact with Lys707, Ser 657, and Gln 658 of TopBP1, whereas the conserved motif Phe, Val, and Pro-Pro (situated -4, -3 and -2, and -1, respectively) can interact with Phe 681, Phe 682, Ala 710, Trp 714, and Met 692 from TopBP1 BRCT5 c2 loop which connects β2 and β3 sheets. Sun et al. also reported the binding of phosphorylated Mediator of DNA damage checkpoint protein 1 (MDC1) with TopBP1 in a special manner. The peptide from MDC1 appears to be trapped between two BRCT4/5 domains and interactions occur in the same amino acids from TopBP1, but with a reversed orientation in the phosphorylated MDC1 peptide [26]. Furthermore, single-BRCT interactions with its phosphorylated target occur with a dissociation constant (Kd) significantly higher than measured for tandem BRCT repeats. These data confirm the interactions of TopBP1 BRCT4/5 with BLM as a monomer and with higher affinity than MDC1 [26]. A similar mechanism is likely involved in other interactions such as TopBP1 with p-53 Binding Protein 1 (53BP1) [27].

Gómez-Cavazos et al. recently described a new BRCT interaction in the binding between the BRCT module from the Epithelial Cell Transforming Sequence 2 (ECT2) and the Rho family Gtpase CYK4 [28]. The basic surface, which is required for hosphor-CYK4 interaction and cytokinesis, is primarily composed of residues from the second domain (BRCT-1) of the triple BRCT module (BRCT-0, 1, 2) of ECT2, supporting the critical role of this single domain in the interactions with phosphorylated CYK4 [28]. Interestingly, this binding mode differs from that observed in tandem BRCT modules where the phosphopeptide binds across the interface formed between the two BRCT domains [23].

The presence of BRCT modules has been noted in many signalling proteins, such as the triple BRCT module present in DNA damage response scaffolding protein TopBP1. These results support the hypothesis that non-canonical interaction modes may mediate BRCT module functions in diverse contexts [28].

With regards to the structure of novel BRCT domains, the tertiary structure of the leishmanial PES BRCT domain was recently calculated via homology modelling [20]. BRCT domains usually show a hydrophobic core of β-sheets trapped between α-helixes with a typical β1, α1, β2, β3, α2, β4, α3 pattern [17] and exhibiting α2 variable among BRCT domains [17,19]. Overall, the constructed models maintained the aforementioned general structure. Furthermore, the hydrophobic motifs present in conventional BRCT domains [18], which would support the typical BRCT folding, were also found in those constructed models. It is well known that such hydrophobic interactions are important for the correct BRCT folding [17].

## 3. Interactions Mediated by BRCT Domains

BRCT domains are DNA/protein binding modules linked to several functions such as DNA repair, cell cycle, or protein interaction, among others. To perform their roles, these domains interact with their partners in both phosphate-dependent and phosphate-independent manners.

### 3.1. Phosphate-Dependent Interactions

The first discovered types of interactions mediated by BRCT domains were phosphate-dependent, as these domains were able to identify the phosphorylation status of their partners. DNA damage response kinases generate specific phosphorylated sequences which can be recognized by BRCT-binding domains. Such interactions then allow signal transduction via the formation of protein-protein complexes at discrete foci within the nucleus [29].

BRCT domains were initially described in the tumour-suppressor protein BRCA1, which featured a pair of tandem BRCT domains extensively studied, and two distinct pockets: one located into the BRCT domain for the interaction with the pSer of the pSer-X-X-Phe peptide motif [29], and another pocket found in the BRCT-BRCT inter region which interacted with the amino acid located at +3 and provided selectivity [19]. Yu et al. demonstrated that the BRCA1 tandem BRCT domain directly bound its partner BRCA1-Associated Carboxyl-terminal Helicase (BACH1) via BACH1 phospho-Ser 990 and allowed BRCA1 to perform DNA damage control at the G2-M checkpoint [30]. Furthermore, BRCT domains from other proteins also preferred phosphorylated molecules. For instance, the BRCT domain of the protein phosphatase Fcp1 primarily interacted with phosphorylated RNA polymerase II. Moreover, the tandem BRCT domains (from Crb2, TopBP1, RAD4, ECT2, DNA Ligase IV, MDC1, and Rad9) and three single BRCT domains (from TDT, REV1, and DNA ligase III) bound more effectively to peptide libraries that exhibited a phosphoserine than to non-phosphorylated peptides [30]. The human TopBP1 has nine BRCT domains and plays a role in DNA replication initiation, checkpoint signaling DNA repair, and transcriptional control [31]. The phosphorylation of Treslin by human CDK enabled TopBP1 to participate in DNA replication initiation. This phenomenon was also described in its yeast homologue Dpb11, another BRCT-bearing protein [31]. In the case of this topoisomerase-binding protein, the tandem BRCT1/2 interacting with BRCT0 (lacking phospho-binding pocket) and the tandem BRCT7/8 have shown a phosphopeptide-binding capacity. However, the single BRCT3 and 6 do not contain the signature amino acids needed for this binding function [31,32], contrary to previous data reported by Yu et al. that highlighted the binding of TopBP1 BRCT6 with the phosphorylated transcription factor E2F1 [30].

To perform this kind of interaction, cooperation between BRCT domains through double and triple tandems is necessary, and such a binding was thought to be conserved among tandem BRCT domains. Nevertheless, more recent discoveries, like the interaction of BRCT5 domain of TopBP1 with phosphorylated MDC1, have revealed other interfaces among tandem BRCT interactions [33]. In this BRCT tandem, only BRCT5 has the phosphate binding pocket [34] which, combined with its structured loop, formed a positively charged surface that bound to phosphorylated Ser-Asp-Thr motif from MDC1. The interaction with phosphorylated MDC1 is required for the recruitment of TopBP1 to stall replication forks [32]. Similarly, although with higher affinity and reversed orientation, BRCT5 from TopBP1 interacted with a phosphoserine from the BLM protein, as recently demonstrated by X-ray crystal structure of the complex [26]. It seemed that BLM phosphorylation and the subsequent interaction with TopBP1 might be responsible for BLM stability [26].

In addition to binding to phosphorylated motifs, BRCT domains, both single and tandem, have also been reported to interact with the free phosphate during DNA breaks. This interaction was speculated to be related to the presence of BRCT domains in proteins involved in DNA damage repair, such as XRCC1 or Rad9, and was first demonstrated using TopBP1 and BRCA1 as model proteins [35]. TopBP1 BRCT1, 1/2, 6, 7, and 7/8, as well as BRCA1 BRCT1/2, were reported to specifically bind nicked DNA oligomers, independently to the tested DNA sequence. In addition, TopBP1 BRCT1, 6, 7/8 could bind to nicked circular DNA as well, and the bindings were independent of the terminal forms (blunt, 3′ protruding and 5′ protruding). Moreover, BRCT domains have also been shown to confer protection against exonuclease digestion [35]. It was clearly demonstrated that DNA polymerase mu (Pol μ) could bind phosphates on dsDNA through a positively charged region generated by three residues (Arg44, Arg52, and Lys54) from its single BRCT domain [36]. Furthermore, the available crystal structures of Replication Factor C (RFC)-dsDNA complex demonstrated and illustrated this kind of interaction [37].

### 3.2. Phosphate-Independent Interactions

Since BRCT domains have been reported as protein-protein binding modules, they may interact differently with BRCTs from other proteins when compared to those interactions that take place in the tandem BRCT folding within the same protein. Some examples of single BRCT interaction have been shown for enzymes involved in Base Excision Repair (BER). The BRCT2 from XRCC1 interacts with the single BRCT from the DNA-Ligase III through salt-bridges and hydrophobic interactions across their α1 helixes [38,39]. The other BRCT from XRCC1, BRCT1, binds to BRCT and to the zinc finger domain from PARP-1 [40,41]. Similarly, the sixth BRCT from TopBP1 interacts with the automodification region of PARP-1, which includes its BRCT domain [42]. Nevertheless, despite the finding of a conserved PARP-binding motif in TopBP1 BRCT6, no PARP-binding capabilities have been shown in vitro [32].

Lastly, BRCT domains are also able to interact with other proteins lacking BRCT. An illustration is the 53BP1-p53 complex generation. The C-terminal region of the 53BP1 containing a BRCT tandem repeat interacts with p53. According to the crystal structure of this protein complex, the residues located in the linker space between the tandem BRCT domain of 53BP1 interact with conserved residues on the loops L2 and L3 of p53 [43].

In summary, to perform their roles in a variety of cellular processes, BRCT bearing proteins interact with both DNA molecules and other proteins with or lacking BRCTs, generating homo or heterodimers. These aforementioned interactions might be phosphate-dependent or phosphate-independent [35,38,43].

## 4. The Role of BRCT in Cell Processes

Proteins, from bacteria to mammals, that bear BRCT domains are mostly implicated in cell processes such as DNA Damage Response (DDR), DNA repair, and/or cell cycle control [13,18].

### 4.1. BRCT in DNA Damage Response

Cellular DNA can be damaged through genotoxic stress leading to the break of DNA strands. These alterations are involved in the development of human pathologies such as malignancies, early aging, or chronic inflammation. Cells have evolved DNA Damage Response (DDR) mechanisms to sense and respond to genome integrity. BRCT domains play a key role in the early stages of this response.

There are two subsets of kinases implicated in DDR: phosphatidylinositol-3-OH-kinase-like kinases (PIKK) kinases and the checkpoint effector kinases (CHK). The BRCT domains, along with Forkhead Associated domains (FHA), are mainly implicated in the recognition of the residues that have been phosphorylated by PIKK [44]. FHA domains select pThr residues whereas BRCT domains tend to select pSer residues [14]. CHK act downstream of the PIKK signalling and phosphorylate serines with an arginine in their -3 position, -XRXXS-, a motif specifically deselected by BRCT domains [44]. Therefore, BRCT domains are not involved in the interaction with motifs phosphorylated by CHKs but act as scaffold for PIKK proteins during the DDR mechanism.

For instance, in chromatin regions bearing DSBs (Double Strand Breaks), proteins belonging to the PIKK family phosphorylate a conserved Ser residue located in the C-terminal of the histone H2AX. The Mediator of DNA damage checkpoint protein 1 (MDC1) binds phosphorylated histone H2AX at DNA damage sites through its tandem BRCT domain [45]. Then, MDC1 is phosphorylated by a serine/threonine kinase (ATM) PIKK-type and the E3 ubiquitin protein ligase (RNF8) joins to this marked protein through its FHA domain, also in a phosphate-dependent manner. Consequently, an accumulation of checkpoint mediator proteins near the damaged chromatin may occur [46]. More specifically, the tandem BRCT of 53BP1 is required for the “slow kinetics” of DSBs repair after radiation-induced DNA damage [47].

On the other hand, DSBs are necessary for the immunoglobulin heavy chain (Igh) class switch recombination (CSR) to take place. There are random mutations into antibody genes that activated B cells carry out through a process of targeted DNA damage and subsequent mutagenic repair. This process needs to be tightly regulated in order to protect the rest of the genome from mutagenesis. Afterwards, the newly generated antibody is able to efficiently bind to its antigen. It was shown that BRCT of PARP-1, a DNA-damage-sensing enzyme which recruits DNA repair enzymes to the sites of the damage [48,49], is required for the mutagenic repair specific to diversifying antibodies. This showed a specialized role of the BRCT domain from PARP-1, contrary to its involvement in high-fidelity DNA repair [50]. Similarly, the C-terminal tandem BRCT domain of BRIT1 (also named MCPH1) participates in the DNA repair phase of CSR through its interaction with phosphorylated H2AX [51].

In summary, BRCT bearing proteins participate in DDR mechanisms by recognizing modules specifically phosphorylated by PIKKs.

### 4.2. BRCT in Cell Cycle Control

Cell cycle checkpoints are involved in the prevention of genetic mistakes during the cell division process. Mutations related to such regulations are linked to different types of cancers, since the affected cells divide continuously and excessively.

Nuclear factor-kappa B (NF-κB) plays a complex role in genome maintenance and tumour progression and suppression by taking part in the regulation of the cell cycle, among others functions. This particular diversity is caused by the structure of NF-κB, which is composed of dissimilar subunits: p50 (NF-κB1), p52 (NF-κB2), p65 (RELA), RELB, and CREL that dimerize in different combinations. For instance, while the overexpression and phosphorylation of p65 is commonly associated with inflammation and cancer, NF-κB1/p50 is usually related to tumour suppression [52]. BRCA1-associated RING domain-1 (BARD1) is an essential protein which dimerizes with BRCA1 and acts as an E3 ubiquitin ligase. It is known that p50 directly links to BARD1 BRCT domains via a C-terminal phospho-serine motif [52]. Such an interaction is induced by p50 S328 phosphorylation performed by the serine/threonine-protein kinase Ataxia telangiectasia and Rad3-related (ATR). This binding to BARD1 induces the mono-ubiquitination of p50 by the BARD1/BRCA1 complex, even though the presence of BRCA1 is not required for the interaction. p50 phosphorylation is increased in the S phase, with the highest levels of phospho-p50 detected in the G1/S transition [52]. Consequently, p50 is mono-ubiquitinated in the S phase. The loss of this post-translational modification increases S phase progression and chromosomal breakage. Genome-wide studies have also revealed a substantial decrease in p50 chromatin enrichment in S phase and Cyclin E as a key factor regulated by p50 during the G1 to S transition [52]. In summary, these results indicate that PTM of p50 at the G1/S transition allows its recognition of a BRCT domain and reduces its binding to the CCNE1 promoter facilitating Cyclin E expression and S phase progression. Furthermore, mutation of p50 increased both Cyclin E and cellular proliferation [52].

Although the unicellular eukaryote *Saccharomyces cerevisiae* seems not to exhibit BRCA1 homolog, the main DNA repair pathways are conserved in humans and yeast [53,54]. Interestingly, the expression of pathogenic BRCA1 variants in yeast induced an increase of homologous recombination (HR) [55]. It is well known that HR is involved in the maintenance of genome integrity repairing DNA double-strand breaks (DSBs) during S and G2 phases of the cell cycle. Moreover, the expression of cancer-related BRCA1 missense variants significantly decrease yeast DNA-damage-induced HR, suggesting a functional interaction between yeast DNA repair and BRCA1.

Recently, Lodovichi et al. performed experiments to express WT BRCA1 and several variants in G1-, S-, and G2-arrested yeast cells and determined the effect on Gene Reversion (GR) [56]. The expression of *WT BRCA1* did not increase GR in dividing and cell cycle–arrested cells, while the pathogenic variants induced a significant increase in GR [56]. Altogether, these results indicate that BRCA1 missense variants are able to induce yeast GR through cell cycle–specific DNA repair pathways [56].

During cell division, BRCT-bearing protein ECT2 activates RhoA in a narrow zone at the cell equator in anaphase [57]. Moreover, Schneid et al. have proven that each ECT2 BRCT domain might critically contribute to ECT2 function [58]. In particular, BRCT0 is linked to, and BRCT1 is essential for, ECT2 activation in anaphase, whereas BRCT2 is involved in two ECT2 functions: GEF inhibition and RACGAP1 binding [58]. Moreover, BRCT2-dependent function of ECT2 limits RhoA activity to the cell equator. In summary, BRCT-dependent signaling of ECT2 activity is required for successful cell division [58].

BACH1 is a protein that belongs to the RecQ DEAH helicase family. Despite being equally expressed throughout the cell cycle, its association with chromatin increases only in S-phase. The regulation of BACH1 helicase activity takes place throughout the interaction with the BRCT repeats of BRCA1 [59].

TopBP1 is another BRCT-bearing protein with key implications in cell-cycle control. *TopBP1* gene expression is induced at the G1/S transition of the cell cycle, and it has been shown to interact with other key mediators of cell cycle, such as MDC1, the E3-ubiquitin ligase HUWE1, and, even if not directly, CK2 [60]. Moreover, TopBP1 function is regulated in a cell cycle–dependent manner. The phosphorylation of some of its partners enables their interaction with the BRCT domains of TopBP1 at specific phases of the cell cycle. TopBP1 is also a target for cell cycle–dependent phosphorylation. Through the phosphorylation of its S1159 by AKT kinase, TopBP1 modifies its binding, stops interacting with Treslin (a key factor for replication initiation in G1 phase), and starts binding to E2F1, which inhibits E2F1-dependent apoptosis as cells progress through the S phase [60]. Treslin interacts with BRCT1-2 in G1, whereas E2F1 interacts with BRCT6 in TopBP1 in S/G2 phase [56].

Additionally, DNA damage checkpoint regulating S-phase entry is controlled by a phosphorylation-dependent interaction of TopBP1 and 53BP1. BRCT4-5 domains of TopBP1 selectively bind conserved phosphorylated 53BP1 [61]. In addition, the interaction between TopBP1 and 53BP1 allows for the formation of RAD9-RAD1-HUS1 complex to the BRCT1 from the same TopBP1 molecule and thus cooperates in ATR activation in the G1 DNA damage checkpoint [61].

BRCTs are also implicated in the repair of DNA lesions that arise during the DNA replication processes in the S phase. For instance, in the fission yeast *Schizosaccharomyces pombe*, the multi-BRCT domain protein Brc1, phylogenetically linked to budding yeast Rtt107 and mammalian PTIP, plays a key role during cell replication [62]. The C-terminal pair of BRCT domains in Brc1 are known to bind phospho-histone H2A (γH2A) at DNA lesions, as well as E3 ubiquitin protein ligase Rhp18/Rad18, which has crucial functions in post-replication repair [62]. In summary, it is now known that yeast protein Brc1 provides scaffolding functions linking γH2A, Rhp18, and Smc5/6 complex to damaged replication forks [62].

### 4.3. Other Biological Roles of BCRT

In addition to its participation in DDR phenomenon, BRCT domains are involved in other cell processes such as those listed below. BRCT-bearing protein BRCA1 has been shown to play a role in the cell cycle–dependent regulation of the metabolism of fatty acids through the binding of phospho-Ser 1263 of Acetyl-Carboxylase 1 (ACC1) to its tandem BRCT domain. Since ACC1 participates in the first step of fatty acid biosynthesis and the up-regulation of such a process is required for carcinogenesis, BRCA1 might act as a tumour suppressor [59]. On the other hand, the BRCT domain of human PES1 has been shown to be required for correct rRNA processing, whereas point mutations of this domain can disrupt the nucleolar localization of this protein [63]. In fact, the BRCT domain from PES1 is essential to the role of this protein in ribosome biogenesis since its truncation blocks the assembly of the PeBoW complex and alters the transformation of the 32S pre-rRNA into mature 28S rRNA [59,64]. Both the BRCT and Bop1 interaction domains are critical for the nucleolar transport of the preribosomal sub-complex Pes1-Bop1 [64]. A BRCT-bearing phosphatase named Fcp1 removes the phosphate group from the C-terminal domain (CTD) of the largest subunit of RNA polymerase II that is involved in mRNA synthesis and processing [65]. Effectively, at the end of the transcription, RNA Pol II enzyme is prepared for the next cycle by dephosphorization. BRCT domains have also been related to cell division process. The GTPase Rho in charge of ring assembly induction to form the two daughter cells is activated by the guanine nucleotide exchange factor (GEF) epithelial cell transforming 2 (ECT2) during anaphase in animal cells [57]. GEF-ECT features three BRCT domains (BRCT-0, -1, and -2) which perform different functions. BRCT-0 and BRCT-1 activate ECT2 in anaphase, while BRCT-2 exhibits a dual role: inhibitory and linker. On one hand, the BRCT-2 domain is required for ECT2 auto inhibition and contributes to successful cytokinesis. On the other hand, BRCT-2 binds to Rac GTPase activating protein (RACGAP1) which, in conjunction with the Mitogen-Activated Protein Kinase Phosphatase 1 (MKLP1), targets ECT2 to the spindle midzone and the equatorial membrane during anaphase [57].

## 5. BRCT Domains in Infectious Agents

BRCT domains also play a role in pathogen biology and during infectious processes caused by viruses, bacteria, fungi, and parasites.

### 5.1. In Bacteria and Viruses

Phylogenetic analyses show that BRCT-bearing proteins (BBPs) are conserved among bacteria and eukaryotes. As mentioned above, the number of BBPs is related to the complexity of the genome. In fact, contrary to eukaryotic cells, most bacteria species seem to have only one or two BBPs: DNA ligase (NAD^+^- and ATP-dependent) and the DNA polymerase III ɛ subunit [16]. The epsilon subunit is part of the catalytic core of DNA topoisomerase III, a holoenzyme responsible for replication in *Escherichia coli*. The interaction of the epsilon subunit with both units (α and θ) forming DNA Pol III has been reported. Besides its alfa subunit-binding domain, the presence of an exonuclease domain positioned at the N-terminal region has been described [66]. It has also been observed that *E. coli* mutants specifically defective in response to nalidixic acid show insertions in the gene encoding the ε subunit of DNA polymerase III, supporting its importance in the quinolone-induced SOS response [67]. In fact, it is known that exonuclease activity of the ɛ subunit can be inhibited by nucleoside 5′-monophosphate such as pNP-TMP [68], supporting its role as a potential target of new-design antimicrobial agents [69].

By catalysing the joining of breaks in the phosphodiester backbone of duplex DNA, DNA ligases play a crucial role in the diverse processes of DNA replication, recombination, and repair. There are two classes of DNA ligases: those which use NAD^+^ as a cofactor, mainly present in eubacteria [70], and those that need ATP as a cofactor which, although ubiquitous, are mainly found in eukaryotic cells and archaea [71]. Since NAD^+^-ligase (LigA) is an essential enzyme in *E. coli*, it is likely that homologous proteins are essential for all bacteria. One region that is particularly well conserved among already highly similar NAD^+^-ligases is the C-terminus BRCT domain, reinforcing the useful and necessary function of this domain. It was shown that, despite not being essential for the ligation activity of LigA, its absence did reduce the efficiency of the in vitro nick-joining reaction, as ΔBRCT-LigA was almost three times less effective at joining nicks compared to the full LigA [72]. Furthermore, it also reduced the rate of double-strand breaks joining in plasmids when compared to full LigA [72]. Similar results were later reported for the same enzyme reported by Wang et al., as they found a significant reduction in enzyme activity after the removal of the BRCT domain [73]. Furthermore, it was shown that such removal also abolished the capacity of *E. coli* LigA to relax supercoiled DNA in the presence of AMP, highlighting the importance of this domain in the formation or maintenance of the AMP-LigA-DNA complex. Moreover, engineered alanine mutations of the BRCT domain were generated and, despite the fact that they reduced LigA activity, their effects were milder when compared to those caused by the removal of the whole BRCT domain. Supercoiled DNA relaxation was also impaired for two of the generated mutants, underlining the essentiality of this domain for processes in *E. coli* and, potentially, in all bacteria [73].

Mimivirus, the largest DNA virus known, also possesses a BRCT-bearing NAD^+^-dependent DNA ligase. The activity of this enzyme is, unlike some bacterial DNA ligases, highly dependent on the BRCT domain. In fact, the BRCT-lacking mutant Mimivirus DNA Ligase shows only 1.3% of the activity of wild-type MimiLIG. In this viral ligase, the BRCT domain seems to be involved in a downstream step of the ligation pathway, since the deletion of a BRCT does not affect the first step adenylation reaction with NAD^+^ [74]. Similarly, the BRCT domain from *Mycobacterium tuberculosis* NAD^+^-dependent ligase is essential for the protein activity and, as with MimiLIG, defects caused by its deletion occur at steps after enzyme adenylation [75].

As previously mentioned, the ATP-dependent DNA ligases are found in bacteria, archaea, and eukarya. Within those bacterial LigA that bear it, the effects of deleting the BRCT domain of proteins range from deleterious to mild or minimal, depending on each enzyme.

For instance, DNA ligIV is an ATP-dependent ligase involved in Non-Homologous End Joining (NHEJ) DNA repair pathway. The presence of certain virus, such as adenoviruses, activates the NHEJ pathway leading viral genome ligation, and consequently the formed concatemers are too large to be packaged into the viral capsid. Adenovirus type 5 is able to redirect the cullin-5 containing RING fingers E3s to ubiquitinate targets that interfere with virus replication, as observed in LigIV through NHEJ pathways. It has been reported that the alfa helix of BRCT-1 from LigIV is needed for such adenovirus-mediated proteasomal degradation [76].

### 5.2. In Parasites

In the human pathogen *Toxoplasma gondii* database, there are at least three putative BRCT-bearing proteins. However, it is still not known if these proteins are functional in the DSB repair pathway of this parasite. Nevertheless, the degree of conservation between yeast and mammalian HRR (homologous recombination repair) BRCT-bearing proteins is very low, despite the fact that their functions overlap [77]. Even if there are no HRR homologs reported for *T. gondii*, considering the low conservation between other organisms and mammals, it is reasonable to assume that the same functions might be carried out by dissimilar BRCT-bearing proteins that share only few residues [77].

In kinetoplastids, a group of flagellated protozoans, the kinetochore is made up of specific proteins which differ from those traditionally detected in eukaryotes. A microtubule-binding kinetochore protein named KKT4 was detected in this group of unicellular flagellated eukaryotes. Contrary to the kinetochore proteins from other eukaryotes, the KKT4 trypanosomatid protein features a phosphopeptide-binding tandem BRCT domain (BRCT1-BRCT2) in its C-terminal [78,79]. BRCT1 interacts with the microtubule-binding domain of the KKT4, as a regulation mechanism of the protein activity, as well as with another kinetochore protein named KKT8 [79]. Similarly, the BRCT domain from the Ras-proximate-1 (Rap1) protein of *Trypanosoma brucei* (TbRap1) is required for its self-interaction [80]. TbRap1 orthologues have been detected from protozoan to mammals and, as with their counterparts, TbRap1 participates in telomere silencing, specifically of variant surface glycoproteins (VSGs) [81] that are used by these parasites to evade the host immune system and perform long-term infections. Among others, TbRap1 bears a BRCT domain useful for normal protein expression levels and, consequently, for proper cell growth. It is thought that self-interaction mediated by the BRCT domain contributes to TbRap1 protection against proteases, thus maintaining normal protein levels [80].

In *L. major*, the BRCT-harbouring protein *LmjPES,* a homolog of the oncogene *PES1,* has been shown to be linked to parasite infectivity. *LmjPES* is demonstrated to be overexpressed in the metacyclic promastigotes, the infective form for vertebrates. In fact, *LmjPES*-overexpressed parasites showed higher infectivity rates and increased their virulence capability in in vivo models. The footpads of mice inoculated with such parasites exhibited higher and faster swelling compared to those of animals infected with wild-type parasites. In addition, an increased induction of iNOS was detected in the inoculation area [82]. Therefore, the BRCT domain from LmjPES has been used as a target when searching for new treatments against leishmaniasis. Using a structure-based drug discovery strategy, a battery of new specific inhibitors targeting BRCT from LmjPES was reported. After in vitro validation, one of such compounds exhibited leishmanicidal activity against promastigotes and amastigotes of three *Leishmania* species without harming macrophages [20]. As previously described, BRCT domains are protein modules involved in cell cycle and DDR, and the relation between mutations in molecules involved in such mechanisms and the development of malignancies has been reported. Links between cancer and infectious diseases, including parasitic pathologies, have also been described. Recently, BRCT from *LmjPES* was expressed in mammalian cells. These transgenic mammalian cells expressing the BRCT domain from *Leishmania* parasites dramatically increased their growth rate and their resistance to DNA-damaging treatments such as etoposide and 5-fluorouracil [83]. In addition, those cells exhibited altered expressions of mitochondrial, proliferation, and chemoresistance genes. In vivo experiments in athymic nude mice revealed a significant capability of those transgenic cells to generate highly proliferative tumours. This study reinforced the existing link between parasitism and cancer development through potential horizontal gene transfer [83]. In addition, it shed some light on the relevance of BRCT domains in parasites.

### 5.3. In Fungi

It has been shown that the transcriptional regulator Rap1p in *Saccharomyces cerevisiae* has a homolog in the human pathogen *Candida albicans*. This yeast protein features a BRCT domain within its N-terminus, which is not essential for *S. cerevisiae* growth [84], but it has been speculated that this region may be the interaction interface between Rap1p and Gcr1p, a transcriptional activator of glycolytic genes [85]. The deletion of both the N-terminus containing BRCT domain of Rap1p from *S. cerevisiae* as well as the complete Rap1p gene in *Candida albicans* can produce hypersensitivity to various cell wall–perturbing agents in those yeasts [86]. This may be due to the relation between the glycolytic pathway and the cell wall thickness [87]. In summary, although the biological significance of the BRCT domain in the Rap1p protein from *S. cerevisiae* and *C. candida* has not been yet elucidated, its conservation among closely related species such as *Kluyveromyces lactis* or *Candida glabrata* suggests some key functions. In addition, the low percentage of identity of these protein domains, including BRCT (24%) [88], from *Candida albicans* Rap1p with respect to those from its ortholog in humans may suggest that such a protein is a potential target for the development of antifungal compounds [89].

## 6. Implication of BRCT Domains in Cancer and Other Pathologic Processes

### 6.1. BRCT Domains and Cancer

It is well known that BRCT domains can be found both as isolated single domains and assembled into complexes with a high variety of protein domains [16]. Accordingly, BRCT-bearing proteins are involved in diverse cell processes, including those related to diseases development. In this section, we review the main involvements of BRCT domains in oncogenic processes.

As described above, BRCT domains were first identified in the breast cancer suppressor BRCA1, a large protein of 1,863 residues with only two small structural motifs: a RING finger domain at the N-terminus, and a tandem of two BRCT domains at C-terminus. BRCA1 has been described as a master regulator of the response to DNA damage by promoting homologous recombination (HR) [90], since its tandem BRCT domains bind to phosphorylated proteins relevant to the cell HR response, particularly Abraxas, BACH1 (BTB and CNC homology 1), and CtIP (CtBP [C-terminal-binding protein]-interacting protein) [91]. The implication of BRCA1 in breast and ovarian cancers is well known [92]. In fact, carriers of germline mutations in BRCA1 showed an increased risk of developing these malignancies, since such mutations appear in 80% of patients with hereditary cancers [93]. BRCA1 exerts its function in response to DNA damage by generating nuclear foci. Gaboriau et al. studied 12 missense mutations, corresponding to different effects on the stability of BRCT domains from BRCA1. Only the mutations affecting the phospho-peptide binding site resulted in a complete inactivation of the protein. In 2019, a study of 78 BRCT missense variants in the UMD-BRCA1 database demonstrated a correlation between the homologous recombination-defective variants, which also present defective BRCT for phosphopeptide binding, and increased cancer risk [94]. On the other hand, some mutations were moderately destabilising since BRCT bearing such variations were able to generate effective foci. It seems that interactions with BRCA1 binding partners help to restore the destabilising effects of some BRCT mutations. The effect of some mutations within the BRCT domain of BRCA1 during the splicing phenomenon has also been reported. Ahlborn et al. showed that some of these variants resulted in splicing aberrations leading to truncated transcripts that can be considered pathogenic, in spite of the fact that most of them did not show protein-level changes during preliminary functional examinations [95].

The correlation between cancer predisposition and nonsense mutations causing a premature stop codon is well known. However, it is difficult to evaluate the effect associated with Variants of Uncertain Significance (VUS), deletions/insertions or intronic variations that preserve the reading frame [55,96]. To predict the possible pathologic implications of VUS in BRCA1, in 2011, Iversen et al. [97] published VarCall, a computational Bayesian hierarchical model to estimate the likelihood of pathogenicity based on results from in vitro functional assays. Afterwards, transcriptional analyses were used to validate functional tests of all VUS in the C-terminal of the BRCA1 protein and validation was completed using yeast-based functional assays for other proteins containing BRCT domains (MCPH1 and MDC1). Therefore, the information from paralogous proteins could also be used to classify VUS [98]. As described above, not all missense mutations of the BRCT domain from BRCA1 alter the activity of the protein with pathogenic results. For instance, a study carried out with two de novo BRCT missense mutations identified in Chinese females with familial breast cancer showed that cells harbouring such variations exhibited similar functions to those lacking them. In fact, the studied mutations reduced growth and increased apoptosis in carrier cells, similarly to those harbouring BRCA1 unmutated. Furthermore, mutants caused S-phase arrest after irradiation and did not sensitize cells to the PARP1 inhibitor Olaparib [99].

BRCT domains from other proteins and their implication in cancer have been studied. It has been shown that tandem BRCT-bearing proteins were related to breast cancer prognosis and staging, as they were less expressed in histopathological samples associated with a worse prognosis. The most statistically significant differences were annotated for PAXIP1 and TP53BP1, which were expressed at low levels in triple negative breast tumours [100] and were early mediators of DDR [100].

BRCT-bearing proteins can also inhibit enzymes involved in cancer development. For example, MCPHI and BRCA1 regulate Telomerase activity. Telomeres are regions of 1000–2000 repetitions of TTAGGG sequences at each end of the chromatin, whose function is to preserve genome stability. They become shorter with each cell cycle. In cancer, stem, and germ cells, telomerase enzyme corrects this shortening process by adding more nucleotide to the DNA ends. One of the subunits of this telomerase enzyme is telomerase reverse transcriptase (hTERT) and its increased activity is related to tumour progression. Alternative splicing of hTERT has been proven to alter telomerase activity, a phenomenon related to diseases like thyroid and gastrointestinal tumours, and myelodysplastic syndromes. As mentioned above, the telomerase activity is also regulated by the tumour-suppressors (BRCT-bearing proteins) MCPHI and BRCA1. The inhibition is mediated by an intact N-terminal BRCT of MCPH1. In fact, BRIT1/MCPH1 act as a negative regulator of the active isoform of hTERT and its expression is reduced in epithelial ovarian cancer (EOC) samples [101].

It is well known that protein domains such as the BRCA1 C-terminal (BRCT) domain mediate in DNA damage response (DDR) signalling events. Moreover, variations in the DDR pathway are linked to certain cancer development. The BRCT domains can exert inhibitory activity on proteins from the DNA damage response pathway. For instance, in cancer cells, the G2 checkpoint has the role of repairing DNA damage. WEE1 is a mitotic protein kinase able to produce G2 phase arrest through CDK1 phosphorylation and, as a consequence, allows DNA reparation in tumour cells. PAXIP1, which contains six BRCT domains organized into three tandem pairs, mainly interacts with WEE1 through its terminal tandem-BRCT. This interaction disrupts the phosphorylation of CDK1 by WEE1, avoiding G2 cell cycle phase arrest and consequently increasing chemotherapy sensitivity. In 2016, Jhuraney et al. demonstrated the synergistic effect between PAXIP1 and WEE1 inhibitor (AZD1775) in lung cancer cells lines. PAXIP1 promoted the progression of damaged cells through the cell cycle, stimulating the anticancer effects of AZD1775 (also known as Adavosertib, MK-1775). In addition, PAXIP1 is linked to apoptosis, and higher levels of cell death were detected when this protein was overexpressed. Although PAXIP1 was not essential for the AZD1775 response, it was clearly involved in the regulation of WEE1 activity at the G2/M checkpoint and it was most likely required for a robust therapeutic response [102].

Another example is the BRCA1-associated RING domain protein 1 (BARD1), which carries a RING finger domain in its N-terminal, as well as three ankyrin repeat domains and a double BRCT domain in its C-terminal region. The BRCT domain from BARD1 is involved in recruiting and retaining the heterodimer BARD1-BRCA1 in DNA damage regions, through its interaction with the Heterochromatin protein (HP1). In 2019, Adamovich et al. studied 76 VUS of BARD1, potentially cancer-associated, and 19 of them tested in the BRCT domain. They found five with reduced Homology Directed Repair (HDR), the cell mechanism involved in repairing double-strand breaks in DNA. Consequently, cells bearing these BARD1-BRCT variants were more sensitive to genotoxic damage caused by cisplatin or ionizing radiation [103]. Besides the double-strand breaks (DSB), DNA damage may also induce Single Strand Breaks (SSB) or relevant base modifications. The XRCC1 protein, which is composed of one N-terminal domain, two BRCTs (BRCT1 and BRCT2), and a Central DNA binding domain (CDB), is implicated in the repair pathways of these two DNA damage processes [104]. To exert such functions, this scaffold protein is able to directly join DNA or interact with PARP through its central BRCT (BRCT1). As a consequence, polymorphisms detected in the DNA-binding residue from BRCT1, which affects its activity, could increase cancer risk [105].

### 6.2. Implication of BRCT in Other Pathologic Processes

BRCT domains are not only related to cancer development but are also linked to other pathological processes. It was demonstrated that the expression of BRCA1 plays a neuroprotective role during brain ischemia/reperfusion (I/R) episodes [106]. This tumour suppressor increased its expression during I/R injuries in brain tissue and conferred a protective effect by diminishing the production of reactive oxygen species (ROS). BRCA1 likely carries out these activities by interacting with NRF2 through its BRCT domain, inducing the nuclear translocation of NRF2 and promoting gene expression of antioxidant response elements (ARE) [106]. BRCA1 is also known to play a protective role in neural stem cells (NSCs), reducing cell apoptosis and promoting proliferation, most likely through blocking p53-mediated apoptosis [107]. Both the RING finger and BRCT domains from BRCA1 are able to bind p53, fostering its ubiquitination and degradation after oxygen-glucose deprivation/reoxygenation (OGD/R) in NSCs. These data shed some light on the interesting possibility of NSCs transplantation after ischemic stroke [107].

Eukaryotic DNA ligases are enzymes that use ATP to realize their activity in DSB repair pathways. However, in 2018, Chen et al. demonstrated that human Ligase IV was shown to recognize NAD^+^ by its tandem BRCT domain, facilitating Ligase adenylation and DNA ligation [108]. Besides mutations of its catalytic site, which are related to DNA Ligase IV syndrome (with severe clinical manifestations such as immunodeficiency, growth retardation, and T-cell leukaemia among others), the pathogenic mutation R814X induces a truncated product that removed the second BRCT domain and avoids its NAD^+^ interaction. Then, ligase IV adenylation is abolished along with its posterior DNA ligation, and consequently, DSB repair is affected [108]. Another example is the gene *microcephalin* (*mcph1*), which is expressed during foetal brain development. It codifies a protein bearing three BRCT domains, two in the C-terminal and one in the N-terminal. The mutations in the *mcph1* gene produce premature chromosome condensation in the early G2 phase of the cell cycle and delayed decondensation postmitosis. In fact, the majority of mutations described so far in the *mcph1* gene result in protein truncation. For instance, missense mutations in both BRCTs from the N-terminus are clinically associated with primary head deformity and human primary microcephaly [109,110]. The function of the BRCT domain from MCPHI in brain size determination was recently studied using a mutant mice model lacking the BRCT from the MCPHI N-terminus. Such mice exhibited reduced brain size and both male and females were infertile, with almost all female mutants developing ovary tumours. In addition, cells harbouring such mutations showed defectiveness in DNA damage response and DNA repair activity, and exhibited the premature chromosome condensation phenotype, coinciding with primary microcephaly patients [111].

Recently, several authors have described the implication of BRCT domains in inflammation. In fact, it was shown that the pharmacological inhibition of PARP, a BRCT-bearing protein, could significantly reduce inflammation induced by LPS in a mouse lung model [112]. As mentioned above, BRCA1 is a BRCT-bearing protein involved in DNA repair, transcriptional regulation, and cell cycle control. The depletion of BRCA1 reduced the expression of pro-inflammatory cytokines TNF-α and IL-6 in cells exposed to LPS. Moreover, BRCA1 knockdown promoted the expression of the anti-inflammatory markers IL-10 and TGF-β [113]. Interestingly, molecular docking analyses predicted interactions between the conserved BRCA-1 domain BRCT and the NF-κB p65 subunit. It was suggested that BRCA-1 depletion might impair pro-inflammatory polarization and activation of RAW 264.7 macrophages in a NF-κB-dependent mechanism [113]. Furthermore, LmjPES, the homologue of the human oncogene PES1, was found in *Leishmania major* and exhibited a BRCT domain [82]. Its gene expression level was highest in the infective forms of the parasite. Additionally, after generating mutant parasites that overexpressed *LmjPES*, a significant footpad inflammation was observed in BALB/c mice infected with those transgenic parasites. Such results highlight the implication of LmjPES, a BRCT-bearing protein, in inflammation [82].

## 7. BRCT as a Therapeutic Target. Novel Techniques for BRCT Domain Research

As mentioned above, BRCT domains take part in a variety of important cell processes including DDR and cell cycle control. In addition, a few of these protein modules have been shown to be involved in pathologies such as cancer or infectious diseases. Therefore, significant efforts have been made towards finding compounds able to specifically inhibit the functions of these protein domains.

To our knowledge, one of the first reports on BRCT-specific inhibitors described the binding of monoacetylcurcumin (Figure 3) to the C-terminal BRCT domain of human DNA polymerase λ, which is involved in an interaction with NHEJ factors to repair the DSB of the DNA [106]. The interactions with such drugs were characterized in silico and predicted to be mediated through connections with residues Thr51, Gly52, Gly54, Ala58, Lys63, Val66, Val85, Glu87, and Ala113 from the BRCT domain. The fact that this curcumin derivative showed no effect on the Terminal deoxynucleotidyl transferase (TdT), a BRCT-bearing protein similar to pol λ that belongs to the DNA polymerase X family, supports the idea that specific-BRCT domains are recognised by this compound [114]. As described above, BRCT protein domains have also been linked to infections including parasitic diseases. Recently, our group described for the first time this domain in *Leishmania* [82]. Furthermore, its structure has been used to search for compounds with leishmanicidal activity. Using a virtual screening method, different compounds were selected based on their interactions with the BRCT domain from *Leishmania major*. Afterwards, these BRCT-interacting chemicals were tested in vitro against parasites. Interestingly, one of them, named CPE2 (Figure 3), showed significant leishmanicidal activity against extra and intracellular forms of the parasite. Moreover, CPE2 did not exhibit toxicity on macrophages and showed a lower action against species causing visceral leishmaniasis. CPE2 activity might be due to key residues in the interaction between this compound and the BRCT domain from *Leishmania* PES protein (LmjPES). CPE2 showed higher interactions with residues present in BRCT from cutaneous leishmaniasis parasites but no interplay was detected with BRCT domains from *L. infantum* and *L. amazonensis*. In addition, these docking key residues are not conserved among BRCT domains from mammal PES proteins [20]. These results might support the previously described observations regarding the interaction between compounds and BRCT domain-specific structures, and might reinforce the role of such protein modules as potential therapeutic targets for the treatment of different pathologies including infectious diseases.

Similar to compounds targeting specific BRCT structures, Bractoppin (Figure 2) is a drug-like BRCT(BRCA1) inhibitor exhibiting the same interactions found in BRCT(BRCA1) by the consensus phosphopeptide, as well as displaying an interaction with the hydrophobic pocket located in the interface between the tandem BRCT fold. Bractoppin was also highly selective for BRCT(BRCA1), since there was no detection of binding to fluorescently labelled BRCT domains from BRIT1/MCPH1, TOPBP1(7/8), ECT2, or TOPBP1(1/2). In vitro, it inhibited substrate recognition by the BRCT (BRCA1) (and not BRCT [MDC1]), with effects similar to those reported for peptide inhibitors (G2 arrest, sensitization to ionizing radiation, and inhibition of HR) [115].

Currently, the search for new drug targets has been spread not only to active sites from enzymes, but also to protein-protein (PPIs) interaction sites. However, unlike enzyme active sites, interfaces involved in PPIs are often large, poorly defined, and may be composed of several contact points. Regarding PPIs, a specific group of amino acids or “hot spots” are responsible for the most important contacts [116]. Therefore, by targeting them, it may be possible to improve the capacity of identifying inhibitors of PPIs [117]. Since hot spot contacts are easy to assay due to the possibility of designing limited-length peptides to mimic one of the interacting partners, high throughput fluorescence assays to discover small molecule inhibitors have been performed [117,118] by using BACH1 binding site on the BRCT of BRCA1 as a target. It was reported that the pSXXF motif, which is the sequence recognized by BRCTs with phosphopeptide-binding function in its target proteins, is the minimal structure for BRCT PPI inhibitors design. In fact, in comparison with longer peptides, pSXXF tetrapeptide showed comparable affinity for BRCT and recreated the most important interactions described for those longer peptides [119]. Regarding the structure of BRCT, it is large and flexible in its unbound conformation [120]. Nonetheless, after the interaction with these PPI inhibitor tetrapeptides, it changed to a more structured state. Most likely, this was due to loops present in the binding interface imposing conformational restraints on the rest of the domain and the network of interactions around pSer and Phe in the pSXXF tetrapeptide, which restricted the dynamical conformation of the unbound BRCT folding. Additionally, BRCT (BRCA1) residues with bigger chemical shifts were either on the binding surface or adjacent to the binding site [120]. Further investigations on this inhibitor were performed by establishing structural minima to achieve nanomolar affinity with BRCT (BRCA1): pSer at position P, a conformational constraint at position P+1, β-branching at the P+2 position, and a salt bridge with Arg1699 at P+3 position [121] (Figure 4). Moreover, a hydrophobic cluster (VLPF) was described as located at P-1 position of the pSXXF binding site, confirming that inhibitors designed to interact with this hydrophobic patch had more affinity for BRCT. In addition, it was suggested that this hydrophobic cluster interaction was conserved in other tandem-BRCT-bearing proteins, such as MDC1 and TopBP1 [121].

Such a peptidomimetic inhibition strategy was first tested in vitro on ovarian and breast cancer cell lines. Cells treated with a BRCA1-BRCT-inhibiting peptide behaved as BRCA1-knockdown cells, blocking its interaction with Abraxas, inhibiting HR, and sensitizing cells to ionizing-radiation (IR) DNA damage [122]. HeLa and MDA-MB-231 cells (but not HCC1937 bearing a truncated BRCA1) treated with the aforementioned inhibitor were sensitive to Olaparib, a PARP inhibitor which has been demonstrated to be effective in BRCA1 and BRCA 2 mutation carriers [122]. Since these mutations were present in a small number of the patients, the use of those new inhibitors may be helpful as adjuvant of Olaparib in patients lacking the mutation. Since peptides were easily degraded and had short half-lives, there was a need for more stable high-affinity binders of BRCT (BRCA1) within the cellular matrix [122]. Accordingly, Na et al. developed a small-molecule microarray (SMM) system to detect PPI inhibitors and found the first cell-permeable BRCA1 BRCT protein interaction inhibitor, which also demonstrated synergism with Olaparib and Etoposide (Topoisomerase II inhibitor) in both in vitro and cell-based experiments [123]. Tumour cells treated with the inhibitor showed more sensitivity to apoptosis induced by ionizing radiation [123].

Most studies have been focused on the design of inhibitors against double-BRCT modules, including those from the BRCA1 protein. However, regarding single BRCT, the existence of small molecule PPI-inhibitors has been also reported. The (±)-gossypol (Figure 2) (especially in its [-] form) is a cell-permeable small molecule PPI inhibitor of PARP1. Most likely, gossypol causes dimerization of the protein by covalently binding imines between its dialdehyde and the residues from the PARP1-BRCT domain, generating an inhibition of the PARP1 enzymatic activity [124].

As described above, a peptidomimetic is a synthetic scaffold that mimics the capability of natural proteins and peptides to recognize a specific target. Moreover, they can be chemically modified to achieve better biochemical and biophysical properties [125]. Since they can be generated either by chemical synthesis or by expression in biological systems, their production is cost-efficient and they are of current interest [126]. For instance, in 2019, Batalha et al. developed cyclic β-hairpin structures that bind selectively to the phosphorylated peptide ligand of the BRCA1-BRCT domain both in silico and in vitro [127], thus enabling the development of new tools to capture and identify BRCA1 BRCT-binding proteins and peptides.

The interaction of BRCT domain of PARP-1 with Ets-1, a transcription factor whose activity is involved in numerous cellular mechanisms, including angiogenesis, apoptosis, and tumour invasion [128], has also been reported. In a later study, using a protein-protein docking in silico method, Brysbaert et al. described BRCT homologs which potentially interacted with the oncoprotein Ets-1 and identified putative binding pairs associated with DNA repair mechanisms [129]. Similarly, the abovementioned drug-like inhibitor Bractoppin was detected via a fluorescence polarization assay, followed by in silico docking and Structure-Activity Relationships [115], reinforcing the advantages of these techniques in drug development.

In 2021, Arshad et al. performed in-silico analyses of 122 non-synonymous SNPs (nsSNPs) of the *BRCA1* gene retrieved from the NCBI SNP database [130]. Using in silico tools, those researchers were able to predict 61 (out of 122) “damaging” nsSNPs. Moreover, they also identified two amino acid residues in the BRCT domain (P1771 and I1707) and five residues in the RING finger domain (L22, C39, H41, C44, and C47) with the potential to have a negative impact on BRCA1 protein function [130]. Those studies provided relevant insights on the usefulness of new in silico techniques to unveil the effects of nsSNPs and amino acid substitutions in BRCT domains including those from BRCA1.

To find inhibitors of PARP-1 through interactions with its BRCT domain, and without interplaying with its catalytic domain, a virtual small molecule screening has been postulated to identify selective inhibitors within PPI hotspot residues (Gly399, Lys400, Leu401, Lys441, and Lys442) and a druggable pocket [131]. Two FDA-approved small molecule drugs (levoleucovorin, a pharmacologically active isomer of racemic leucovorin, and balsalazide, the anti-inflammatory drug used in the treatment of inflammatory bowel disease) that are already in trials for cancer treatment may apparently fit in the druggable pocket.

In a report by You et al., who searched for inhibitors of the BRCT PPI through molecular dynamics (MD), the thermodynamic advantage of flexible peptide ligands when compared with rigid compounds or their rigidized counterparts is highlighted, emphasizing a novel strategy for the design of compounds able to inhibit a flexible domain such as BRCT [132].

## 8. Conclusions and Future Perspectives

BRCT domains have been identified in a wide group of living organisms and viruses. Considering their relevant implications in key biological processes (Figure 5) related to DNA repair and replication, the role of BRCT domains in human cancer development has been assessed extensively. Nevertheless, due to their structural plasticity and presence in other organisms, future research is required to study their potential pharmacological exploitation in cancer and other diseases including Neglected Tropical Diseases (NTDs). Those pathologies, such as leishmaniasis or African trypanosomiasis, cause significant social and economic burdens and yet they still lack treatment with effective and selective drugs.

Therefore, novel in silico techniques, such as molecular docking and machine learning, are useful tools to achieve those goals. Based on the structural bioinformatics of BRCT domains, the available ligand structures and features, robust classification and prediction schemes for the diagnosis of diseases, and the design of therapeutic compounds might be proposed. In addition, synergies may be explored for the simultaneous use of BRCT ligands with other modulators, such as PARP catalytic inhibitors.

This review may contribute to a better understanding of BRCT domains and their possible implications for innovative drug development. BRCT-bearing proteins are pharmacological targets, which would allow new treatment opportunities against a wide range of diseases, including NTDs.

## Figures and Tables

**Figure 1 pharmaceutics-15-01839-f001:**
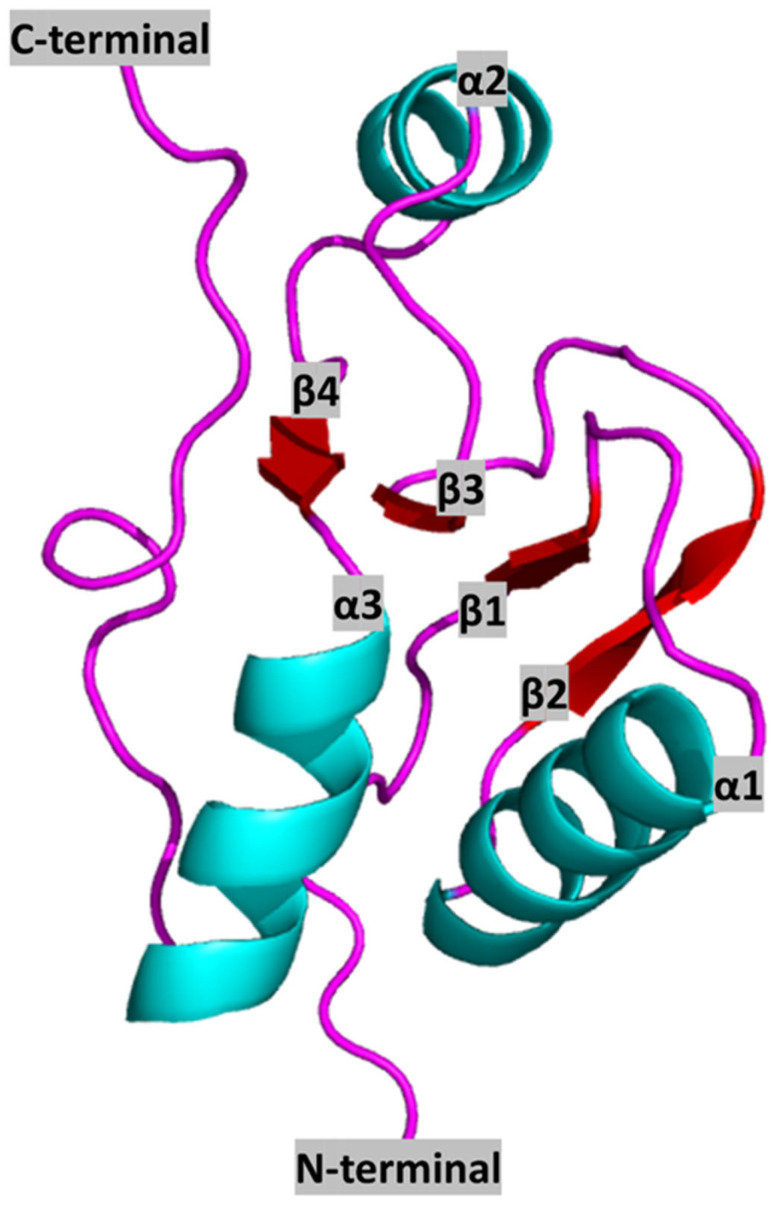
Representation of a canonical BRCT domain (from human XRCC1 protein) (PDB ID: 1CDZ) [17]. Colours represent secondary structure elements: red: β-sheets; cyan: α-helixes; pink: Loops. N and C terminals, as well as secondary structure elements, are annotated in black letters.

**Figure 2 pharmaceutics-15-01839-f002:**
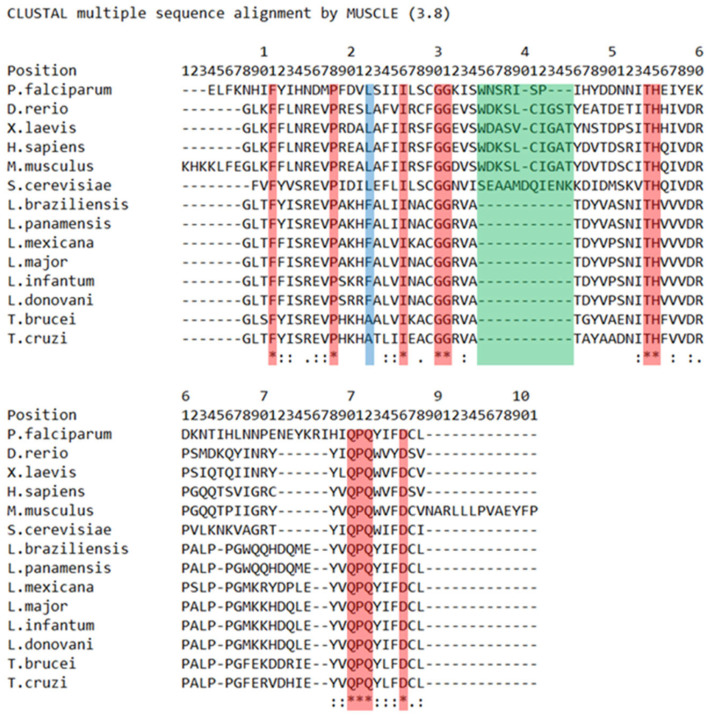
PES1 BRCT domain protein sequence is highly conserved in trypanosomatids and moderately conserved among others organisms. *L. major* PES BRCT and selected orthologous proteins sequence alignment. Asterisks (*) (red tint) mark positions with single fully conserved amino acids. High (colon [:]) or weak (stop [.]) conservation between amino acids is also indicated. Blue tint: Trypanosomatid distinctive amino acid. Green tint: Trypanosomatid distinctive gap. (Unpublished, adapted from Peña-Guerrero, J et al. [20].).

**Figure 3 pharmaceutics-15-01839-f003:**
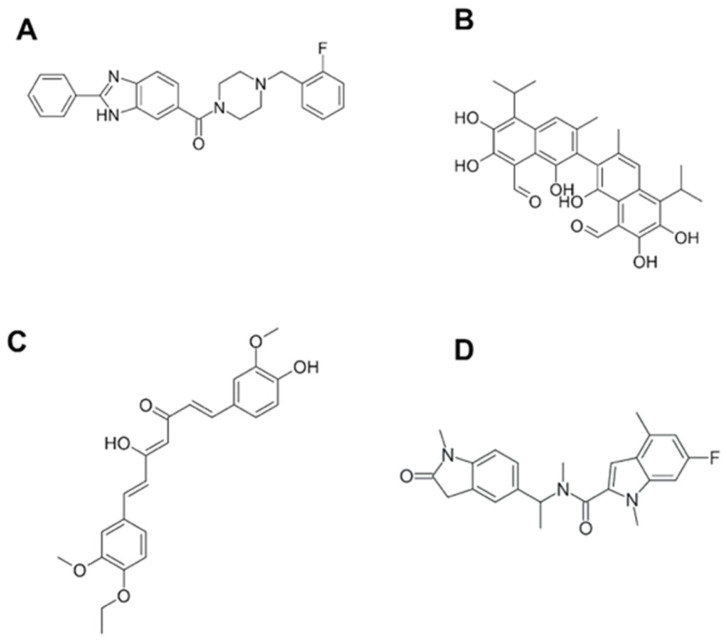
Structures for some of the known BRCT drug-like inhibitors. (**A**) Bractoppin. (**B**) Gossypol. (**C**) Monoacetylcurcumin. (**D**) CPE2.

**Figure 4 pharmaceutics-15-01839-f004:**
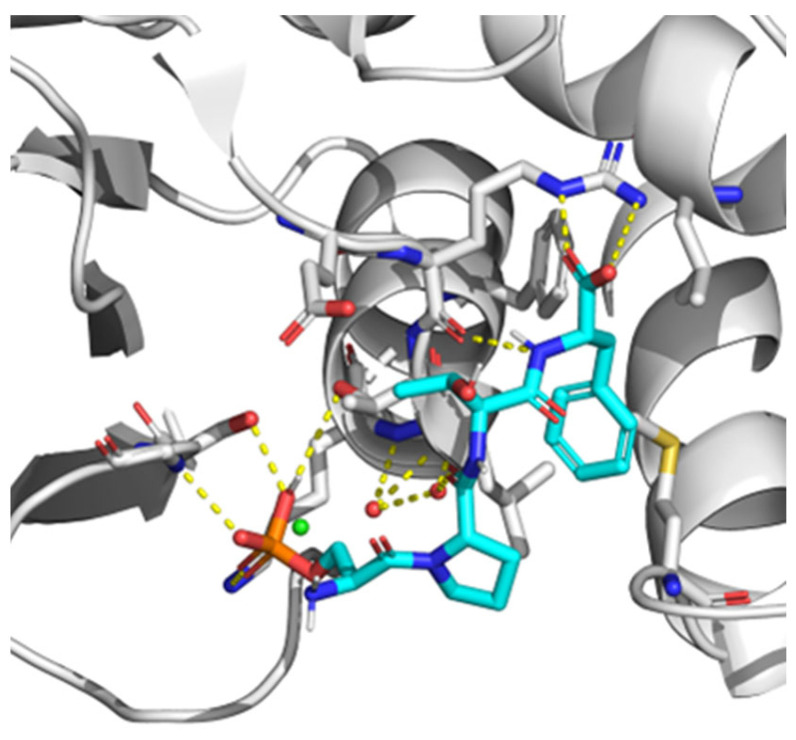
X-ray crystal structure of BRCA1 BCRT domain interacting with minimal recognition tetrapeptide ligand pSPTF-COOH. White: Protein carbon atoms. Cyan: Ligand carbon atoms. Blue: Nitrogen atoms. Red: Oxygen atoms. Green: Chlorine atoms. Orange: Phosphorous atoms. Yellow (dashed lines): Hydrogen bonds.

**Figure 5 pharmaceutics-15-01839-f005:**
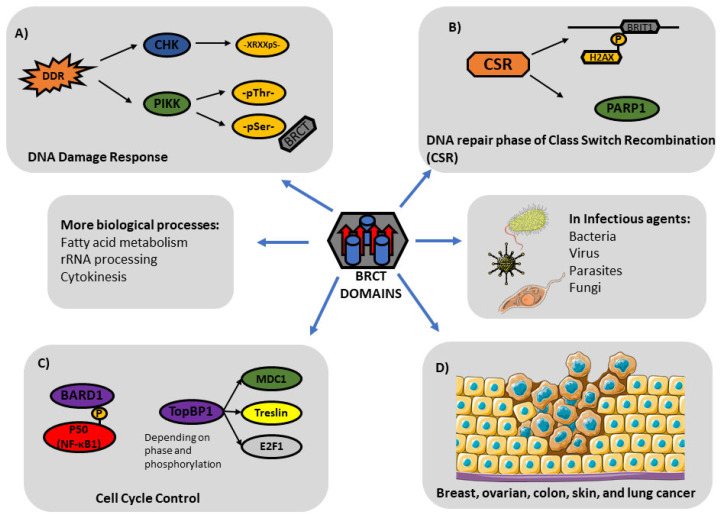
BRCT domains (harbouring 4 β-sheets and 3 α-helixes) play critical roles in biological processes. (**A**) DNA Damage Response (DDR): BRCT domains typically act downstream of PIKK Kinases during DDR, recognizing phosphoSer residues. (**B**) DNA repair phase of Class Switch Recombination (CSR): PARP-1 and BRIT1 (also named MCPH1) participate in the DNA repair phase of CSR. (**C**) Cell Cycle Control: Phosphorylated P50 interacts with BARD1 for cell cycle control. TopBP1 is another BRCT-bearing protein with key implications in cell-cycle control. (**D**) BRCT domains are also involved in diverse cellular processes in cancer and other pathologies.

## Data Availability

Not applicable.

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
