# Peer review of "BRCT Domains: Structure, Functions, and Implications in Disease—New Therapeutic Targets for Innovative Drug Discovery against Infections"

_pharmaceutics, 2023, doi:10.3390/pharmaceutics15071839_

Round 1
Reviewer 1 Report
This review is well-written and clearly summarizes the structure and role of BRCT domain. I have only a few comments.
It would be better to describe change of function after BRCT domain binds to specific proteins in phosphate-dependent interactions. In other words, authors should describe how phosphorylation is affected by the interaction with BRCT domain.
It is interesting if authors add descriptions of implication of BRCT domains in inflammation of other organs such as skin, colon and lung.
Reviewer 2 Report
In the review entitled “BRCT Domains: Structure, Functions, and Implications in Disease. New Therapeutic Targets for Innovative Drug Discovery Against Infections”, the authors explored the possible roles of BRCT domains as therapeutic targets for drug discovery. They described their common structural features, relevant interactions and pathways, as well as their implications in pathologic processes. Drugs commonly used to target these domains were also presented. Finally, based on their structures, they described new drug design possibilities by using modern and innovative techniques.
This review in interesting in order to shed light upon mechanisms that may be harnessed to fight cancer and infectious diseases.
The review is clear, simple to read and well organized in paragraphs.
My only suggestion is to add some figures and tables, related to the main paragraphs, in order to better outline the contents of each topic they decided to achieve.
In particular, it would be interesting to add some figures and tables in which the authors show main pathways and genes regulated by BRCT elements, in order to shed light on the key role of this elements on cell fate and main regulated pathways.
I think this review may be considered for publicaton after these slight revisions.
No comments
Reviewer 3 Report
This review article is exploring the role of BRCT domains, including its structure functions and implications in many diseases. Article is covering almost all data available up today on possible roles of BRCT domains as potential target for drug discovery.
The manuscript is decorated with 4 figures, describing BRCT structure and architecture, protein sequences and structure of known BRCT drug-like inhibitors, and X-ray crystal structure of BRCA1 BCRT domain. Review is also concluded with 130 references. Additionally, article is highlighting the role of BRCT domains in human cancer and other diseases including Neglected Tropical Diseases (NTDs). This will constitute the important goals and novelty of this manuscript!
The following suggested changes and recommendations should be introduced before the publication of the manuscript.
1. Page 1, line 38, replace “works” with “data”.
2. Page 2, line 53, replace “works” with “data”.
3. Page 2, line 67, replace “studied” with “analyzed”.
4. Page 2, line 70, replace “work ” with “ review”
5. Page 5, line 70, replace “way of” with “ specific”
6. Page 7, line 242, insert “ clearly” before “demonstrated”
7. Page 7, line 285, replace “avoid” with “are not involved in“
8. Page 8, line 317, insert “particular” before “ diversity”
9. Page 9, line 391, insert “such as listed below” after “processes”
10. Page 11, line 448, replace “from” with “reported by”
11. Page 11, line 476, replace “need: with “needed”
12. Page 12, line 537, replace “sum” with “summary”
13. Page 15, line 692, correct Figure 2 to Figure 3.
14. Page 15, line 706, correct Figure 2 to Figure 3.
15. Page 18, line 823, replace “Years ago” with “It has been reported by”. Insert “who: after “You et al.”
16. Page 19, line 844. The sentence “In summary.. “is not specific and should be replaced by short two sentences paragraph describing real conclusive summary!!
The manuscript is of good quality and importance and is relatively well written and edited in order to meet the standard for the articles published in Pharmaceuticals. Thus, I certainly recommend it for publication after the correction of these suggested minor changes.
